# Mapping and phasing of structural variation in patient genomes using nanopore sequencing

Mircea Cretu Stancu[1], Markus J. van Roosmalen[1], Ivo Renkens[1], Marleen M. Nieboer[1], Sjors Middelkamp[1], Joep de Ligt [1], Giulia Pregno[2], Daniela Giachino [2], Giorgia Mandrile[2], Jose Espejo Valle-Inclan[1], Jerome Korzelius[1], Ewart de Bruijn[1], Edwin Cuppen[3], Michael E. Talkowski[4,5,6], Tobias Marschall [7,8], Jeroen de Ridder[1] & Wigard P. Kloosterman[1]

Despite improvements in genomics technology, the detection of structural variants (SVs) from short-read sequencing still poses challenges, particularly for complex variation. Here we analyse the genomes of two patients with congenital abnormalities using the MinION nanopore sequencer and a novel computational pipeline—NanoSV. We demonstrate that nanopore long reads are superior to short reads with regard to detection of de novo chromothripsis rearrangements. The long reads also enable efficient phasing of genetic variations, which we leveraged to determine the parental origin of all de novo chromothripsis breakpoints and to resolve the structure of these complex rearrangements. Additionally, genome-wide surveillance of inherited SVs reveals novel variants, missed in short-read data sets, a large proportion of which are retrotransposon insertions. We provide a first exploration of patient genome sequencing with a nanopore sequencer and demonstrate the value of long-read sequencing in mapping and phasing of SVs for both clinical and research applications.

[1] Department of Genetics, Center for Molecular Medicine, University Medical Center Utrecht, Utrecht University, 3584 CG Utrecht, The Netherlands. [2] Medical Genetics Unit, Department of Clinical and Biological Sciences, University of Torino, Orbassano 10043, Italy. [3] Department of Genetics and Cancer Genomics, Center for Molecular Medicine, University Medical Center Utrecht, Utrecht University, 3584 CG Utrecht, The Netherlands. [4] Center for Genomic Medicine, Massachusetts General Hospital, Boston, MA 02114, USA. [5] Department of Neurology, Harvard Medical School, Boston, MA 02115, USA. [6] Program in Population and Medical Genetics and Stanley Center for Psychiatric Research, The Broad Institute of M.I.T. and Harvard, Cambridge, MA 02142, USA. [7] Center for Bioinformatics, Saarland University, 66123 Saarbrücken, Germany. [8] Max Planck Institute for Informatics, 66123 Saarbrücken, Germany. Mircea Cretu Stancu and Markus J. van Roosmalen contributed equally to this work. Correspondence and requests for materials should be addressed to W.P.K. (email: w.kloosterman@umcutrecht.nl)

Second-generation DNA sequencing has become an essential technology for research and diagnosis of human genetic disease. Sequencing of human exomes has resulted in dramatic increases in novel gene discovery for Mendelian disorders[1], while whole-genome sequencing has revealed that a myriad of diseases are caused by genetic changes that can occur both within genes as well as in the noncoding genome[2]. As a result, genome sequencing has seen rapid adoption in clinical decision making, as the complete picture of a patient's unique mutation profile enables personalization of treatment strategies[3,4].

Robust methods to detect structural variants (SVs) in human genomes are essential, as SVs represent an important class of genetic variation that accounts for a far greater number of variable bases than single nucleotide variations (SNVs)[5]. Moreover, SVs have been implicated in a wide range of genetic disorders[6].

A particularly revolutionary development in genome sequencing is the use of protein nanopores to measure DNA sequence directly and in real time[7]. The first successful implementation of this principle in a consumer device was achieved in 2014 by Oxford Nanopore Technologies (ONT) with the introduction of the MinION[8]. The MinION can sequence stretches of DNA of up to hundreds of kilobases in length, which already resulted in the sequencing of the genomes of several organisms[9,10]. Because MinION-based sequencing requires almost no capital investment and current devices have a very small footprint, mainstream adoption of these sequencers has the potential to fundamentally change the current paradigm of sequencing in centralized centers.

An important and natural application of the long reads produced by nanopore sequencing is identifying SVs. Long-read sequencing is breaking ground for the discovery of SVs at an unprecedented scale and depth[11]. The first success has been achieved using the Pacific BioSciences SMRT long-read sequencing platform[12,13], and alternative methods to capture long-range information have been introduced, such as BioNano optical mapping[14] and 10× Genomics linked-read technology[15]. While short-read next-generation sequencing data rely on multiple (often) indirect sources of information in order to accurately identify SVs, structural changes can be directly reflected in long-read data.

In this work, we demonstrate sequencing of the whole diploid human genomes of two patients on the MinION sequencer at 11–16× depth of coverage. The two patients suffer from congenital disease resulting from complex chromothripsis. We employ a novel computational pipeline to demonstrate the feasibility of using MinION reads to detect de novo complex SV breakpoints, at high sensitivity. The long reads from the MinION allow efficient phasing of genetic variations (SNVs as well as SVs) and enable us to resolve the long-range structure of the chromothripsis in the patients. Moreover, we identify a significant proportion of SVs that are not detected in short-read Illumina sequencing data of the same patient genomes.

## Results

**Sequencing of patient genomes with nanopore MinION.** As a first step toward real-time clinical genome sequencing, we evaluated the use of the MinION device to sequence the genomes of two patients with multiple congenital abnormalities[16], henceforth denoted as Patient 1 and Patient 2, respectively.

We extracted DNA from patient cells and sequenced this on the MinION. For Patient 1, we used R7, R9, and R9.4 pore chemistries (Supplementary Table 1) generating a total of 8.2M template sequencing reads from 122 sequencing runs. For Patient 2, we exclusively used R9.4 runs and performed only 13 runs (1.89M reads), which required ~5 days of sequencing on seven parallel MinION instruments at a cost of around $7000

(Supplementary Fig. 1), and produced a coverage of 11×. We observed that 82.1% (Patient 1) and 98.9% (Patient 2) of these reads could be mapped to the human reference genome and were useful for further analyses. Read lengths were highly variable for Patient 1, as a result of differences in library prep methods, with a mean of 6.9 kb for template reads, while for Patient 2 we reached an average of 16.2 kb with consistent read-length distributions across each of the 13 runs (Supplementary Fig. 2).

Raw sequencing data were transformed into FASTQ format using Poretools and sequence reads were mapped to the human reference genome (GRCh37) using LAST[17]. We uniquely aligned 99% of R7/R9 2D reads or R9.4 1D reads flagged as "passed" after EPI2ME base calling, while this dropped to 55% for R9-based "failed" 2D reads (Supplementary Fig. 3). We evaluated the mapping accuracy by calculating the percentages of identical bases (PID) between mapped reads and the reference genome. We observed a mean PID of 90% for R7 2D and R9 2D, 85% for R9 template and 89% for R9.4 template reads based on LAST mapping (Supplementary Fig. 4). An analysis of error rates and types, within the Patient 2 data (i.e., R9.4 reads only), shows that from an observed per-base error rate of 15.1%, indel errors were the dominant error class (10%: 9.1% deletions, 0.9% insertions), followed by mismatches (5.1%). We found a 2.6-fold increase in deletion errors for sequences overlapping homopolymers and 1.4-fold for sequences overlapping tandem repeats (Supplementary Fig. 5). Furthermore, both deletion and mismatch rates were increased in regions with high GC content (Supplementary Fig. 6).

We obtained a mean coverage depth of 16× and 11× for Patient 1 and Patient 2, respectively (Supplementary Fig. 7). Coverage was lower in regions with higher GC content, yet this effect was much less pronounced than for the Illumina sequencing data of the same genomes (Supplementary Fig. 8)[12]. This finding was confirmed by analysis of k-mer distributions of MinION data (Supplementary Fig. 9). We note that while the MinION reads marked as "fail" show systematic sequencing biases regarding coverage distribution, the quality of the aligned fraction is comparable to the "pass" reads. We therefore included the "fail" data of Patient 1 that was successfully retrieved through alignment, in all subsequent analyses.

**Resolving de novo genomic rearrangements with long-read data.** Both patients have complex phenotypes involving dysmorphic features and mental retardation, likely caused by their de novo complex chromosomal rearrangements, which were karyotypically defined as 46,XX,ins(2;9)(q24.3;p22.1p24.3)dn (Patient 1) and 46,XY,t(1;9;5)(complex)dn (Patient 2)[16].

We evaluated the performance to detect the breakpoints underlying the complex de novo karyotypes of Patient 1 and Patient 2 using MinION sequencing data, at this medium coverage. Both patients have already been described in recent work, in which Illumina sequencing was used to map the rearrangement breakpoints, as the current gold-standard method for routine genome-wide SV mapping in patient genomes[16,18]. For Patient 1, we augmented the previously described data by performing Illumina HiSeq X data for both parents. We performed SV calling with Delly[19] and Manta[20] on the Illumina data from Patient 1 and its parents. By integrating SV calls from Delly and Manta and removing calls that were also identified in one or both parents, we obtained a set of 44 putative de novo SV breakpoints, 40 of which formed a complex genomic rearrangement, as described previously[16]. These 40 breakpoints were verified by orthogonal breakpoint assays using PCR and MiSeq sequencing (Supplementary Table 2). The breakpoints cluster within regions of chromosomes 2, 7, 8, and 9 and are the result of

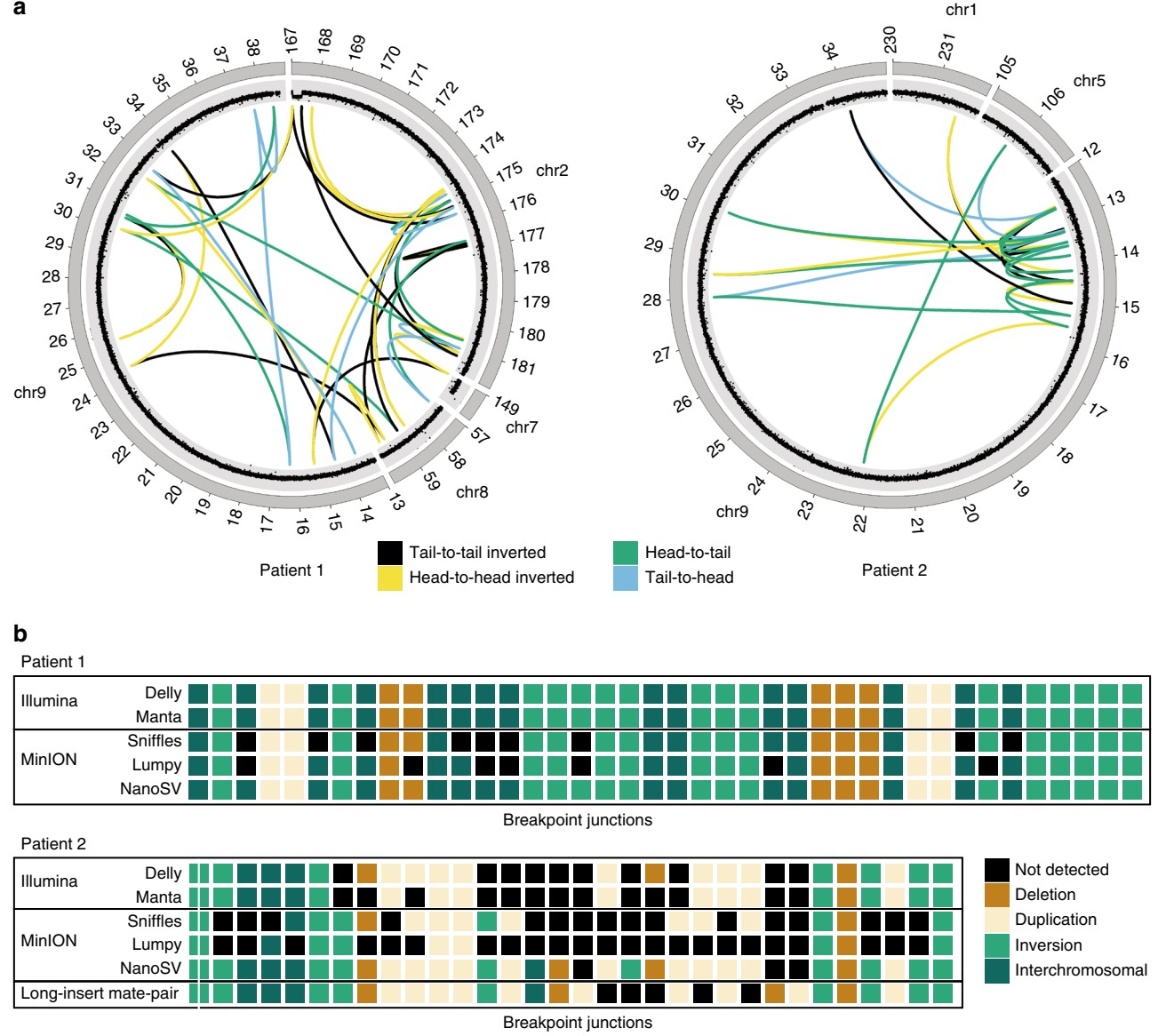

**Fig. 1** Chromothriptic de novo breakpoint junctions of Patient 1 and Patient 2. **a** Circos plots for Patient 1 and Patient 2, respectively. For Patient 1, we took the set of 40 validated de novo breakpoint junctions obtained by Illumina whole-genome sequencing of the patient and its parents. For Patient 2, we depicted the breakpoint junctions as published recently[16]. The outer ring of the circos plot shows the chromosomes and the inner ring shows the copy number changes as revealed by FREEC[34] analysis of Illumina whole-genome sequencing data for both patients. Colored lines (arcs) indicate breakpoint junctions. **b** SV genotyping comparison across the chromothriptic breakpoint junctions, between Illumina Hiseq data and MinION data, using various tools tested. The x-axis represents different breakpoint junctions and the y-axis shows different SV calling methods and data sets. The individual breakpoint junctions are indicated by colors specifying the type of breakpoint junction

a complex shattering and reassembly process, known as chromothripsis[21,22] (Fig. 1a).

For Patient 2, there were 29 SVs underlying the complex de novo karyotype as based on the previously described breakpoint junctions, which were detected using long-insert mate-pair sequencing and revealed a complex chromothriptic rearrangement involving chromosomes 1, 5, and 9 (Fig. 1a and Supplementary Table 2)[16].

To enable SV detection in nanopore long-read sequencing data, we developed a new bioinformatic tool, NanoSV, tailored to nanopore data. NanoSV uses split read mapping (obtained from LAST alignment) as a basis for SV discovery ("Methods" section and Supplementary Fig. 10), and supports discovery of all defined types of SVs (Supplementary Fig. 11). The performance of NanoSV was first evaluated on simulated

nanopore long-read data of an artificially rearranged chromosome and benchmarked against two other recently published SV callers, Lumpy[23] and Sniffles[24]. We thus generated 501 simulated rearrangement breakpoints on chromosome 1 and generated equal amounts of simulated nanopore reads of the rearranged, as well as the reference chromosome, using NanoSim[25] ("Methods"). We assessed performance of NanoSV, Lumpy, and Sniffles on these simulated data with varying read coverage (1× to 44×). We observed that NanoSV reaches 99.2% recall at 27× coverage (Supplementary Fig. 12) with a maximum false positive rate of 1.2% (at 44× coverage). For Lumpy and Sniffles, we reached maximum recall rates of 92.4 and 92.6%, respectively (at 44× coverage) and maximum false positive rates of 78.8% and 97%, respectively.

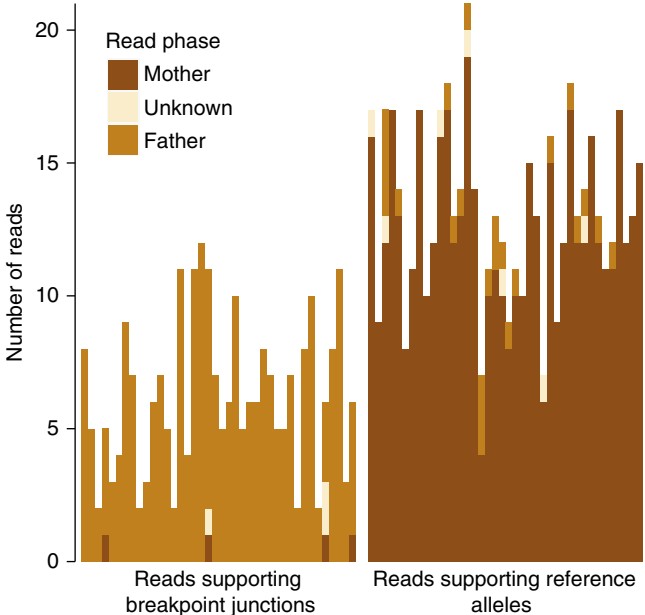

**Fig. 2** Phasing of chromothripsis breakpoint junctions. Phasing of MinION reads overlapping 40 chromothripsis breakpoint junctions in Patient 1. The x-axis displays each of 40 chromothripsis breakpoint junctions identified in Patient 1, stratified by allele (alternative and reference). On the left side only reads supporting the alternative allele are depicted and on the right side reads supporting the reference allele are shown. The y-axis indicates the number of reads supporting each allele, for each of the 40 breakpoint junctions. Legend colors indicate whether the assigned read phase was paternal, maternal, or unknown

We went on to apply NanoSV to the complex chromosomal rearrangements data of our patient genomes, and compared results again against Lumpy[23] and Sniffles[24] for the MinION data and Manta[20] and Delly[19] for the corresponding Illumina data. For Patient 1, we identified 100% of the 40 validated breakpoint junctions. Conversely, we discovered 33 (83%) and 31 (78%) of the 40 de novo breakpoint junction in the call sets from Lumpy and Sniffles, respectively (Fig. 1b). For Patient 2, NanoSV detected 24 of the 29 previously described breakpoint junctions. We investigated further why five variants were missed, using Sanger sequencing of PCR products of the respective breakpoint junctions. We found that two out of the five previously published breakpoint junctions represent a complex combination of more than two joined segments (Supplementary Fig. 13 and Supplementary Table 2). These short segments were not detectable at the lower resolution of long-insert jumping libraries that were used in the previous analyses compared to the long-read capabilities of MinION sequencing used here[16]. Based on validation by Sanger sequencing, we retrieved a total of 32 chromothripsis breakpoint junctions in Patient 2 and 29 (91%) of these were detected using NanoSV (Fig. 1b). For the three remaining breakpoint junctions, insufficient nanopore read coverage hampered proper genotyping. Nevertheless, for the reads that did span these breakpoints, split read mappings supporting each of these junctions were observed. Lumpy and Sniffles detected 9 (28%) and 16 (50%) of the 32 breakpoints junctions in the nanopore data from Patient 2, respectively; Manta and Delly detected 19 (59%) and 22 (69%) of the 32 breakpoint junctions, respectively, in the short-insert Illumina data of Patient 2. To assess the effect of sequence coverage on breakpoint junction detection in real data, we subsampled the Patient 1 data. This produced an estimate of ~14× for the minimum coverage needed to detect all chromothriptic breakpoint junctions (Supplementary Fig. 14).

**Unraveling the long-range structure of chromothripsis**. It has been suggested that germline chromothripsis originates from paternal chromosomes[21], but this has previously been inferred from only a few breakpoint junction sequences or deleted segments. A thorough validation of the conjecture that the origin of chromothripsis is exclusively paternal is lacking. Furthermore, the structure of the chromothripsis rearrangements is typically inferred from the patterns of breakpoint junctions, under the assumption that the chromothripsis breakpoint junctions occur on a single haplotype[21,22,26].

We developed a bioinformatic pipeline to augment genome-wide genetic SNP phasing with nanopore read-based phasing of SVs ("Methods"). In a first step, we obtained 1.7M heterozygous SNPs from Patient 1 that were called from Illumina sequencing data and trio-phased using GATK PBT[27] and Patient 1's parents' genotypes. Subsequently, each nanopore read was assigned phase based on a phasing score that takes into account the content and number of overlapping phase-informative SNPs ("Methods"). Per chromothripsis breakpoint junction, we obtained between 2 and 11 break-supporting nanopore reads and 85% (195/228) of these overlapped on average of 9.8 phase-informative heterozygous SNPs. We similarly phased the nanopore reads that spanned but did not support the breakpoint junctions (i.e, reference reads). This analysis demonstrated that all 40 de novo chromothripsis breakpoint junctions are of paternal origin (Fig. 2). A few breakpoint supporting reads point to an origin of some chromothripsis breakpoints on maternal chromosomes. However, these are all reads with three or less overlapping phase-informative SNVs, and likely represent artifacts. These results support earlier hypotheses of a paternal origin of germline chromothripsis, pointing to a breakage and repair process specific for male chromosomes occurring either during spermatogenesis or early zygotic cell divisions[21]. We were further able to reconstruct the affected derivative chromosomes of Patient 1 by following the chain(s) of breakpoint junctions by order and orientation (Fig. 3a, b). Such a strategy leads to a configuration of four derivative chromosomes for Patient 1, each containing one centromere and two telomeric chromosome ends. The chromosomal structure obtained by this procedure matched the observed karyotype (Supplementary Fig. 15).

We further sought to investigate the extent to which the derived chromosomal structure could be reconstructed from the MinION sequencing data. We note that a much higher sequencing depth is required in order to accurately reconstruct such large chromothriptic regions through diploid assembly. In order to evaluate the potential of nanopore long-read data to facilitate future analyses, we pre-phased, as described above, all the reads that align within the chromothriptic region (i.e., ~40 MB of genomic sequence across four chromosomes) and used only the reads that are known to originate from the paternal haplotype and those that could not be assigned phase (i.e., where the two haplotypes were identical).

We first built contigs by evaluating the read overlaps from the reference alignment ("Methods") and obtained contigs that connect between two to five chromothriptic segments, spanning up to 2 MB of contiguous DNA sequence (Fig. 3c and Supplementary Fig. 16). Finally, we used Miniasm[28] to evaluate whether such longer, local haplotype structure can be readily retrieved in a standardized and scalable fashion ("Methods"). The whole 40 MB region was assembled into 178 contigs that were subsequently aligned against the human reference genome. We identified three contigs of 241 kb, 469 kb, and 1217 kb in size, each spanning three to five chromothriptic segments. Segment order and orientation in each of the three contigs supports the predicted chromothripsis structure (Fig. 3d and Supplementary Fig. 17).

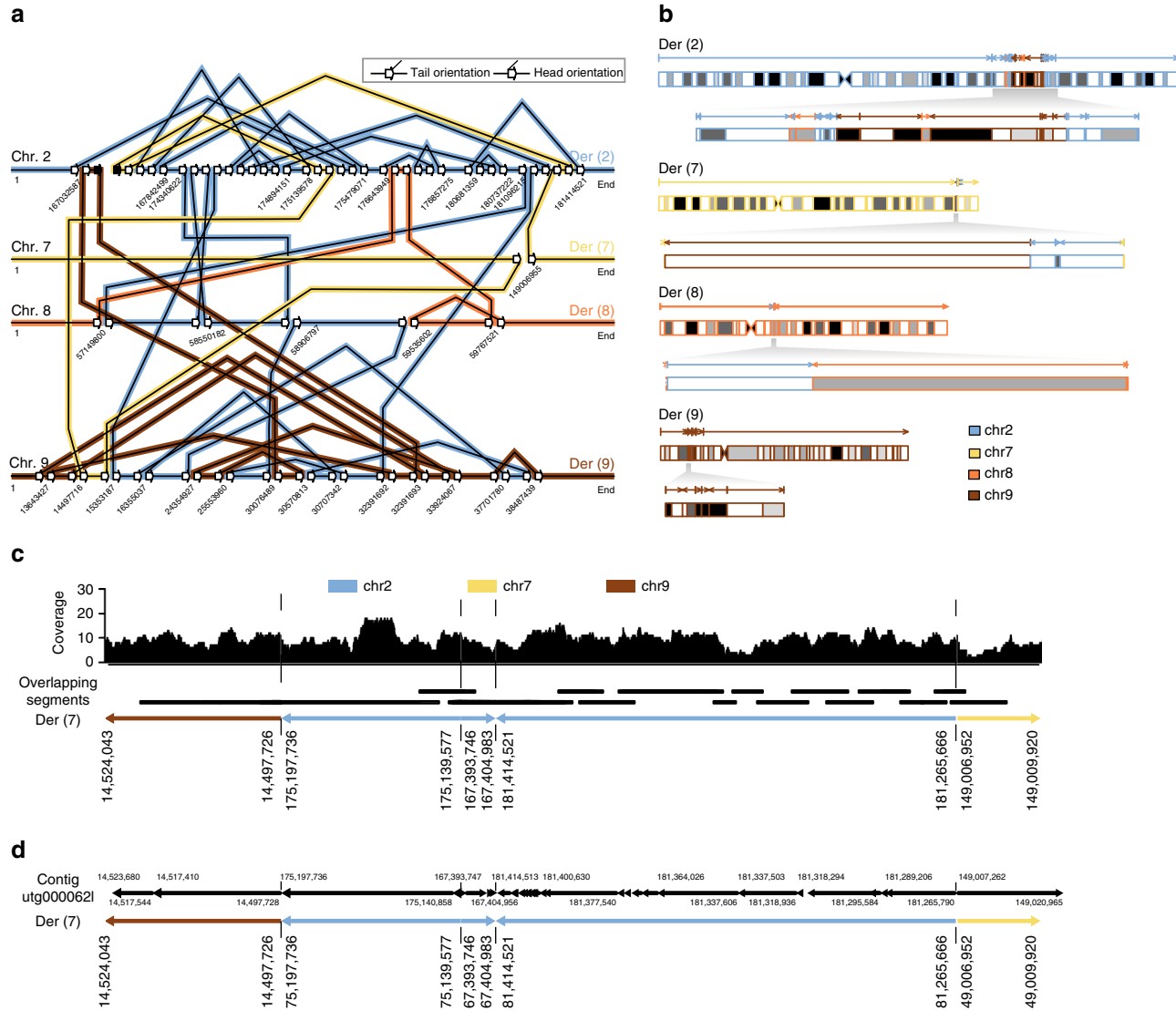

**Fig. 3** Unraveling long-range chromothripsis structure from the nanopore data. **a** Schematic diagram showing the patterns of breakpoint junctions in Patient 1. The human reference genomic regions that are involved in the chromothriptic event are depicted horizontally for each affected chromosome. The slanted lines connecting various reference segments represent breakpoint junctions. The orientations of breakpoint junctions are indicated by arrows as shown in the legend. Black (instead of open) arrows indicate the boundaries of a chromosomal deletion resulting from the chromothripsis, whereas open arrows indicate double-stranded DNA breaks. **b** Structure of the chromothriptic derivative chromosomes in Patient 1, as inferred from the orientations and order of breakpoint junctions shown in **a**. **c** Reconstruction of a chromothriptic subregion of chromosome 7, involving five chromosomal segments. Overlapping aligned reads originating from Patient 1's paternal haplotype were used. Nanopore reads that are instrumental for segment connectivity are indicated by black bars. The coverage track has been generated from all paternal reads mapping to the respective chromosomal segments. The underlying derived chromosome's structure is illustrated on the bottom. **d** Haploid assembly results of the chromothriptic region of Patient 1. A 469 kb contiguous assembled sequence (utg000062l) spans, through 54 segments that align back to the reference genome, the same chromothripsis subregion illustrated in **c**. The assembled contig is fragmented into many (54) aligned segments, as Miniasm does not compute a consensus sequence

**Evaluation of SV calling in NA12878 nanopore data**. Beyond detection of specific pathogenic SVs, long sequence reads present unique advantages for SV discovery in human genomes, as it has been recently shown from data generated on Pacific Biosciences platforms[12,29]. Here, we assessed whether MinION sequencing data could yield comprehensive and high-quality sets of genome-wide SV calls, as well as whether it may yield any novel SVs beyond those found through the Illumina sequencing. To evaluate the performance of NanoSV in a genome-wide analysis, we used the publicly available MinION data for the NA12878 sample[30] and publicly available sets of SV calls, for the same sample, both from short-read Illumina data[31] (referred to as 1KG), as well as from Pacific Biosciences data[13] (referred to as PB). Based on these

calls, we carried out an assessment of sensitivity, as well as accuracy of our analyses.

We aligned all the fastq MinION R9.4 reads that were generated using normal ONT library preparation, for the NA12878 sample (i.e., we did not include ultra-long read data available for NA12878[32]). We then restricted the analysis to chromosome 1 (as a representative subset) and used NanoSV to produce an initial set of 3957 genotyped SV calls. Manual inspection of SV candidates within the NA12878 sample as well as within our patients' data showed that MinION sequencing and base calling performs poorly in regions containing homopolymer stretches, which typically lead to a collapse of the whole region into a spurious indel call. This is observed across samples, as well

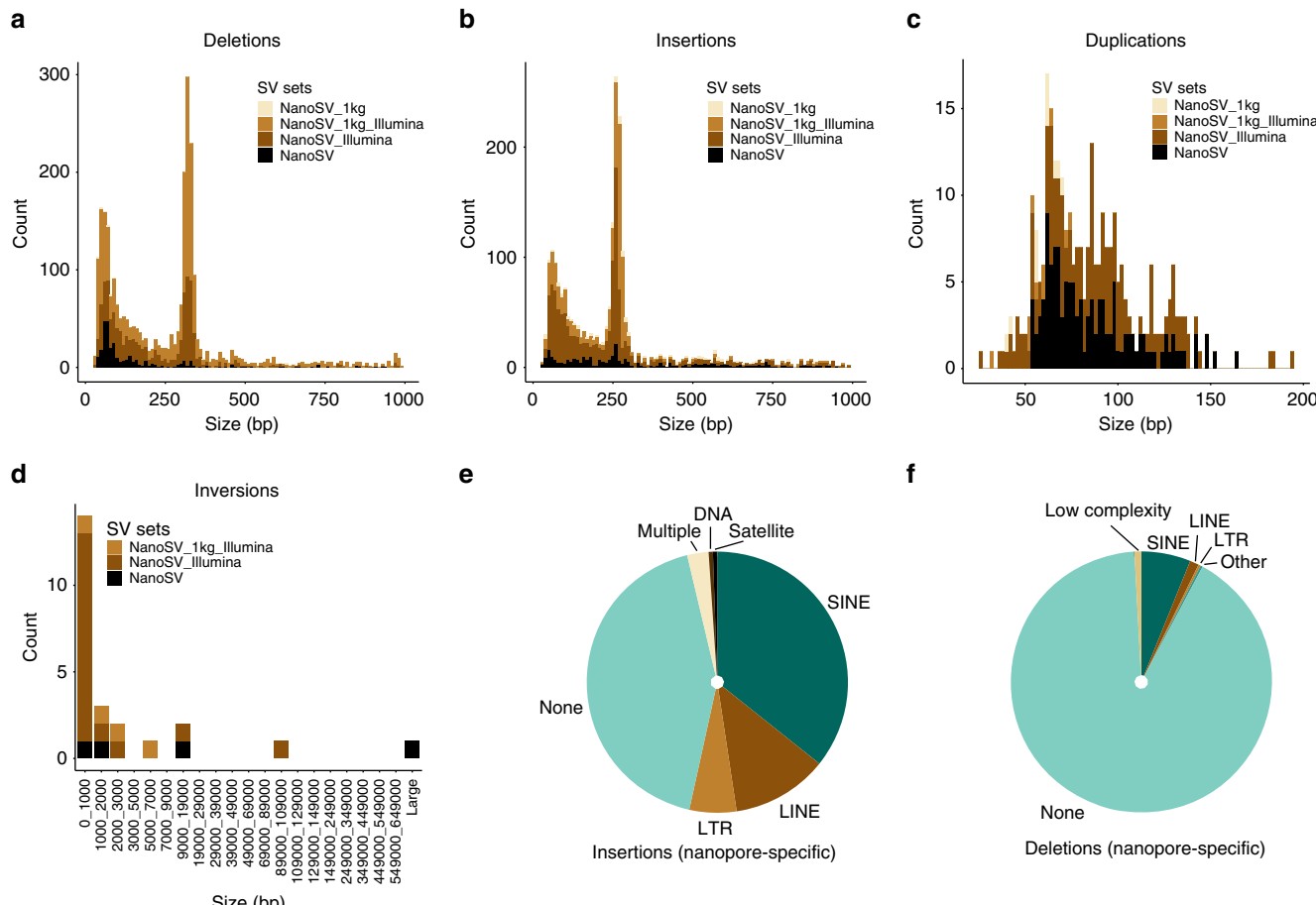

**Fig. 4** Genome-wide detection of SVs using nanopore sequencing data. **a–d** The total amount of high-confidence NanoSV calls for Patient 1 and Patient 2 jointly, across different SV size bins and stratified by SV type as follows: **a** deletions, **b** insertions, **c** duplications, and **d** inversions. The NanoSV calls were intersected with SV calls from other data sources (Illumina data of Patient 1 and Patient 2 and 1KG phase 3 sites). For **a**, **b**, the x-axis was trimmed to 1000 bp for visibility and a small number of variants beyond this size are not displayed in the figure. Similarly, for **c**, the x-axis was limited to 200 bp. **e** The repeat content of nanopore-specific insertions. **f** The repeat content of nanopore-specific deletions. Repeat annotation was obtained from the UCSC repeat masker table (GRCh37)

as in MinION sequencing of PCR products (Supplementary Fig. 18). Additionally, we noted that SV calling is similarly hampered in tandem repeat regions (Supplementary Fig. 18). Based on these observations, we conservatively discarded calls for which both ends of the candidate breakpoint junction fall within genomic homopolymer regions or short tandem repeat stretches, resulting in a set of 657 SVs in NA12878. We further filtered for small indels (<40 base pairs) that do not typically result in a split alignment, resulting in a final set of 654 SVs from the chromosome 1 nanopore data of NA12878. We ran Lumpy and Sniffles on the same NA12878 nanopore data and filtered the resulting SV sets, as well as the gold standard truth sets (1KG and PB) using the same criteria, so as to enable an informative comparison. After intersecting the NanoSV call set with 1KG and PB ("Methods"), we observed a sensitivity of 78% (131 out of 168 1KG SVs) and 88% (292 out of 332 PB SVs), respectively. The largest proportion (18/37) of the SV calls that were missed in the comparison to 1KG are multiallelic CNVs, which typically require dedicated coverage analysis and are absent from the PB data set as well. We further missed six indels that were close to the threshold for creating a split read (i.e., 40–50 bp). Identical evaluations of Lumpy and Sniffles revealed sensitivities of 15% (26/168) and 72% (121/168) in the 1KG set of SVs, and 32% (105/332) and 77% (255/332), respectively, in the PB data set. We note that

Lumpy was designed and tested on short-read paired-end sequencing data and we used it on long-read data as the algorithm is conceptually applicable, rather than specifically tailored.

For all subsequent analyses of the NA12878 sample, we considered all the SVs also preset in the 1KG or PB data sets as true positive SV calls (TPs) and any additional SV calls made by NanoSV as false positive calls (FPs). Out of our set of 654 NanoSV calls, 354 overlap with an SV call in the 1KG or PB data sets, resulting in an estimated precision of 54%. Similarly, Lumpy and Sniffles show precisions of 2 and 50%, respectively.

To further improve post-calling filtering, we trained a random forest model that produces a high-confidence set of SVs, with a precision beyond the 54% mark. The features based on which the model is trained are extracted from the aligned sequencing data and are designed to be sequencing read-depth and read-length independent, such that the model is applicable to any MinION sequencing setting ("Methods"). We select as optimal, a random forest model with 82% precision and 75% sensitivity, on our training data (Supplementary Fig. 19). The data used for training are the 354 TP and 300 FP NA12878 SV calls described above.

**Genome-wide SV discovery from MinION reads**. We went on to analyse the whole-genome MinION data of Patient 1 and

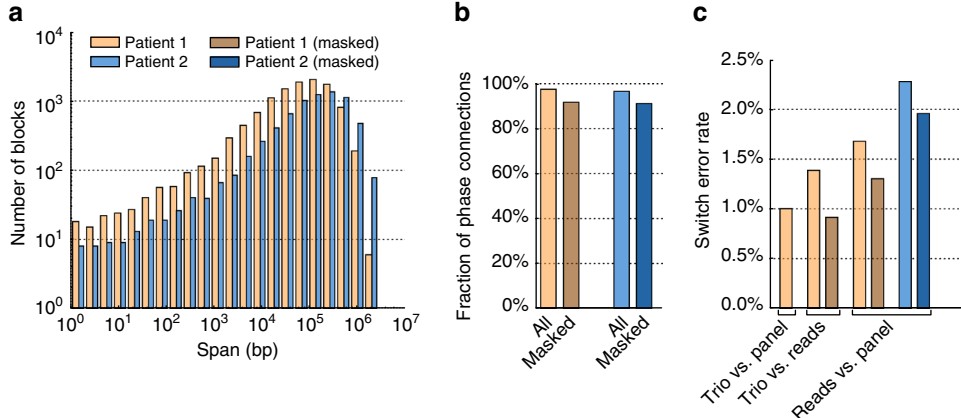

**Fig. 5** Performance of SNV phasing using nanopore reads. **a** Distribution of phased block lengths resulting from read-based phasing by WhatsHap. Patient 1 and Patient 2 are shown in brown and blue, respectively. **b** Fraction of phase connections (i.e., pairs of consecutive SNVs phased with respect to each other) established in the two patients and with/without masking repeats (light/dark colors). **c** For Patient 1, switch error rates of all pairs of trio-based (PBT), population-based (ShapeIt), and read-based (WhatsHap) phasing are shown. For Patient 2, where no family data is available, read-based phasing is compared to population-based phasing

Patient 2. We ran NanoSV and obtained initial call sets of 36,959 and 36,321 SVs, respectively. Filtering for all SVs that do not overlap homopolymers or simple repeats, we obtain 8578 and 6791 SVs in Patient 1 and Patient 2, respectively. Finally, we ran the random forest model trained on the NA12878 data, as described above, and obtained final call sets of 3271 and 3345 SVs, for Patient 1 and Patient 2, respectively.

To further evaluate the robustness of our analysis pipeline, we performed multiple rounds of orthogonal validation, on a random sample, spanning all SV classes and size ranges ("Methods"). We obtained validation status for 274 SVs, regardless of the random forest prediction outcome, for Patient 1, and 77 SVs predicted as true by the random forest, for Patient 2. Based on these sets, we obtained precision estimates of 95 and 96% for Patient 1 and Patient 2 and a sensitivity estimate of 72% for Patient 1.

We intersected the SV call sets of Patient 1 and Patient 2 with calls generated by Lumpy and/or Sniffles. Furthermore, we performed SV calling on the corresponding Illumina data of both patients using six tools (Pindel, Manta, Delly, FREEC, Mobster, and GATK HaplotypeCaller) that are commonly used in human genome sequencing studies and which represent different methods to detect SVs (and/or indels) from whole-genome short-read Illumina sequencing data, that collectively capture most classes of SVs[19,20,33–35]. An SV is considered to be overlapping with the Illumina data set if the nanopore data SV call matches an SV call in any of the tools used on the Illumina data ("Methods"). We further considered as overlapping Illumina data (i.e., "detectable" through short-read sequencing) any NanoSV-called variant that can be matched within the 1KG SV and indel sites, respectively (Supplementary Fig. 20)[36]. Finally, we annotated the SVs from both patients for overlapping repeat elements from the UCSC repeat masker track or the DFAM database ("Methods").

We identified 14% (944) of SVs in Patient 1 and Patient 2 nanopore data that were not observed in Illumina data nor are they 1KG variant sites (Fig. 4). A comparison of the two sets of SV calls shows that nanopore-specific SVs are on average located at sites with a higher GC content (i.e., than SVs also genotyped from Illumina data), which are typically hard to sequence with short-read technologies (Supplementary Fig. 21). The most frequent class of SVs in the set of 6616 predicted true positive SVs are deletions (54%), of which 10% (360) are novel variants

detected by nanopore data (Fig. 4a and Supplementary Fig. 22). We observed that SINE elements were proportionally less abundant among nanopore-specific deletions (6 vs. 30% among calls overlapping with Illumina data, Fig. 4f). The major fraction (91%) of nanopore-specific deletions is not overlapping a repeat feature, most likely due to our very stringent initial filtering of simple repeats. In fact, the majority (66%) of the nanopore-specific deletions are smaller than 200 bp, while only 27% of all deletions are smaller than 200 bp. Short deletions are known to be hard to detect using short-read sequencing[37]. Insertions represent the largest fraction among the nanopore-specific set of variants (382, Fig. 4b). We observed a proportional increase in the amount of LINEs among nanopore-specific insertions compared to calls overlapping Illumina data (12 vs. 8%), while SINEs are proportionally underrepresented in nanopore-specific insertions (36 vs. 42%) (Fig. 4e and Supplementary Fig. 22). Finally, 41% of all detected (tandem) duplications (337) are novel variants detected by nanopore data (Fig. 4c).

**MinION read-based phasing of SNVs.** Phasing genetic variation is critical for human disease studies[38,39]. To demonstrate the potential of long-read nanopore sequencing data for direct read-based phasing of genetic variation, we employed WhatsHap, an algorithm that we recently established[40,41]. Using WhatsHap, we phased a set of high-quality genome-wide SNVs from both patients ("Methods") and obtained haplo-blocks with N50 = 126 kb for Patient 1 and N50 = 305 kb for Patient 2, respectively. The distribution of block lengths is shown in Fig. 5a. We were able to establish 97.5% (96.5%) of all possible phase connections in Patient 1 (Patient 2), where a phase connection is defined as the relative phase between two consecutive heterozygous SNVs (Fig. 5b). For Patient 1, where Illumina sequencing data were available for the parents, we produced a ground-truth phasing by genetic haplotyping, that is, by using the SNV genotypes and the family relationship[27]. Additionally, we phased both samples using ShapeIt2 and the 1KG phase 3 reference panel[31]. Figure 5c shows pairwise comparisons of the obtained haplotypes, with switch error rates of 1.7 and 2.3% when comparing read-based and population-based phasing for Patient 1 and Patient 2, respectively. We observed a lower switch error rate of 1.4% between trio-based and read-based phasing, which points to a significant amount of switch errors in the population-based phasing (1.0%

when comparing trio-based vs. population-based phasing). Therefore, a significant amount of differences between read-based and population-based phasing is most likely due to errors in the population-based phasing. Since MinION reads are especially prone to errors in homopolymer regions, we investigated the effect of excluding all SNVs in such regions from phasing (see "Methods" section for a precise definition). This resulted in a decrease in the number of established phase connections from 97.5 to 91.7% for Patient 1 and from 96.5 to 91.1% for Patient 2 (Fig. 5b) and a decrease in the switch error rate with respect to the pedigree-based phasing from 1.4 to 0.9% in Patient 1, see Fig. 5c. This shows that switch errors are indeed often found at such homopolymer sites and that masking those sites significantly reduces switch error rates at the expense of only a moderate reduction of phased variants.

**MinION read-based phasing of SVs.** While structural variation has recently been integrated in larger population genetic reference panels, which enables their imputation for genetic association studies[18,36], building these panels often requires statistical phasing approaches, which drop accuracy for low allele frequency SV sites. Read-based phasing of SVs using long reads will enhance our ability to include SVs in high-quality reference panels, where structural variation is still underrepresented[18].

We apply the same methodology as above (i.e., used for phasing chromothriptic breakpoints) to evaluate genome-wide SV phasing accuracy. A total of 3.8M MinION reads overlapped one or more of the 1.7M genome-wide phase-informative SNPs. As estimated from reads overlapping at least 20 phase-informative SNPs, an average of 85.2% of the SNPs spanned by a read consistently support a particular phase assignment, which is in line with the reported error rate of MinION sequencing data (Supplementary Fig. 23). A distinction between reads originating from paternal or maternal haplotypes can be readily made, particularly if reads overlap with multiple phase-informative SNPs (Supplementary Fig. 24). We then selected a set of 2389 heterozygous SVs that overlap between Manta (Illumina) and NanoSV (nanopore) call sets. Each SV was assigned a phase and a phasing quality ("Methods"), by combining information from all phase-informative SNPs falling within the breakpoint junction supporting reads and reference supporting reads, respectively. In this way, we phased 1909 (78.7%) SVs and could assign 971 and 938 to paternal and maternal chromosomes, respectively. For the remainder of 480 SVs, spanning reads did not overlap any phase-informative SNP and therefore a phase could not be assigned to these SVs. Using the SV phasing produced by PBT as ground truth, our long-read-based phasing of SVs had an accuracy of 98.5%.

## Discussion

In this work, we show the first standalone analysis of MinION whole-genome sequencing data of human, diploid, patients' genomes, demonstrating the feasibility of long-read sequencing of human genomes on the MinION real-time portable nanopore sequencer. Given the long-read nature of the MinION platform, we focused the analysis on the detection of clinically relevant SVs, a diverse category of genetic variation that is often causal to human genetic disease[42]. Hundreds to thousands of such patients are routinely screened annually for pathogenic SVs in clinical genetic centers, most often by copy number profiling or karyotyping. Although these methods are robust and relatively cost-efficient, they are not capable of mapping small or copy-balanced SVs, nor do they provide base-pair resolution accuracy, or the possibility to resolve complex SVs[43].

Here we show that MinION sequencing provides an attractive alternative approach for genome-wide detection of clinically relevant SVs, which could be implemented as a clinical screening tool for patients with congenital phenotypes, such as intellectual disability[44]. We developed a robust SV discovery and genotyping pipeline that can produce SV calls matching any state of the art precision benchmark (>95% precision). Due to the medium coverage, some intrinsic nanopore sequencing biases and for benchmarking purposes, we employ extremely stringent filtering that results in a good estimated sensitivity (~75%), which can be further increased through higher sequencing depth, or by relaxing our post-calling filtering steps.

We were able to extract all known de novo breakpoint junctions for Patient 1 (Fig. 1), even at relatively low coverage (16×). The long reads identified additional complexity for several breakpoint junctions of Patient 2. Moreover, 32% (29 vs. 22) more chromothripsis breakpoint junctions were detected with MinION compared to short-insert Illumina sequencing. Our work also supports previous results that revealed many novel SVs and indels discovered from PB long-read sequencing of haploid human cells[12,29]. Through our standalone, genome-wide analysis of SVs in (diploid) patient genomes, we show that long-read nanopore data can be readily applied to any research question for which SVs may play a role. We observed that 14% of the high-confidence set of SVs in the nanopore data could not be found in matching Illumina sequencing data (despite extensive variant calling using six different variant calling methods as well as comparison to 1KG variant sites). Although this percentage of novel variants is lower than for previously reported PB data (89%), this is partly due to our conservative SV calling and post-calling filtering steps. Long MinION sequencing reads thus enable a straightforward and homogeneous analysis of SVs, while retaining a very high accuracy in the final set of variants.

Phasing of genotyped SVs—relevant for mapping disease associations—is commonly done using statistical methods or by employing family relationships among sequenced individuals[18]. We here devised a computational strategy that allowed accurate phasing of SVs directly from the long nanopore reads using flanking (phased) heterozygous SNPs. Read-based phasing of SVs is advantageous particularly for classes of SVs with a low population frequency and for de novo variations. This is exemplified by the evidence provided here for the paternal origin of all de novo breakpoint junctions in Patient 1, whereas previous work on chromothripsis has not provided robust support for the parental origin of chromothripsis[21].

If MinION/ONT data quality and throughput increase at a similar pace as we have observed recently (Supplementary Figs. 1 and 4), SNV calling and genotyping may be directly performed based on the nanopore reads. Even though our data are of relatively low coverage, we were already able to obtain a good genotype concordance (96%) with the Illumina-based pipeline, for existing SNV calls in Patient 1 (not further investigated here). SNV calling combined with accurate genome-wide phasing, as we demonstrated here, will enable simultaneous long-read-only genetic variation discovery and phasing.

Long sequencing reads facilitate personal genome assemblies and emerging new ways of dealing with genetic variation discovery and representation[45,46]. Efforts to obtain full-length haplotype resolved chromosomal sequences are continuously advancing and the combination of multiple long-range sequencing and mapping approaches have recently led to diploid human genome assemblies with contig N50 size of well over 10MB[13,47]. We have not attempted a full human genome assembly using the MinION reads in this work (primarily due to insufficient coverage). However, we were able to separate reads by haplotype, which formed the basis for a reconstruction of the long-range

structure of chromothripsis rearrangements. Such information is essential for interpretation of clinical phenotypes[48].

A drawback of current short-read genome sequencing technology is the need for high capital investment, which often leads to sequencing infrastructure being located in dedicated sequencing centers. This is associated with a complex logistic workflow and relatively long turnaround times. Our results show that such limitations can be overcome by the use of the portable MinION sequencing technology. Since the start of this project in April 2016, we have seen a tenfold increase in throughput per MinION sequencing run (Supplementary Fig. 1) and an increase in sequencing quality to 90% accuracy for high output 1D runs (450b/s). In practice, this means that 10× coverage of the human genome can be reached using 10–15 MinION flowcells at a cost of $5000 to $8000 within 1 week of overall sequencing time.

This work demonstrates the potential of long-read, portable sequencing technology for human genomics research and clinical applications. Creating larger catalogs of SVs, in complex repeat regions and segmental duplications, is a particular challenge in the coming years. We foresee that population-scale genome sequencing by ONT or other long-read technology will facilitate such discoveries, leading to further understanding of the role of SVs in the human genome in general and in genetic disease in particular.

## Methods

**Sample source.** The DNA for human genome sequencing in this study was obtained from two patients with congenital abnormalities and the parents of one of them. Informed consent for genome sequencing and publication of the results was obtained from all subjects or their legal representatives. The study was approved by Institutional Review Boards of San Luigi University Hospital and Brigham and Women's Hospital and Massachusetts General Hospital. Both patients have been previously described by Redin et al.[16].

**DNA extraction.** DNA of Patient 1 was obtained from either peripheral blood mononuclear cells (PBMCs) derived from blood and from renal epithelial cells obtained from urine. Renal cells were cultured up to eight passages as reported previously[49]. Cells were harvested after reaching confluency by trypsinization with TrypLE Select (Thermo Fisher Scientific) and centrifugation at 250×g for 5 min. DNA from the parents was obtained from PBMCs. PBMCs were collected by a ficoll gradient. In brief, the blood was diluted 4× with phosphate-buffered saline (PBS). Subsequently 13 mL of Histopaque®-1077 (family 1; Sigma-Aldrich 10,771–500ML) was added per 35 mL of diluted blood. The resulting mixture was centrifuged at room temperature for 20 min at 900×g, followed by recovery of the PBMC layer. PBMCs were washed twice using PBS, centrifuged at 750×g for 5 min and resuspended in PBS with 50% DMSO. For Patient 2, DNA was obtained from a lymphoblastoid cell line, which has not been tested for mycoplasma contamination. The cell line was authenticated by whole-genome sequencing. DNA extraction from cultured cells and PBMCs was performed using DNAeasy (Qiagen) or Genomic-tip (Qiagen) according to the manufacturer's specifications with exclusion of vortexing to maintain DNA integrity.

**MinION library preparation and sequencing.** Isolated DNA was sheared to ~10–20 kb fragments using G-tubes (Covaris). Subsequently, genomic libraries were prepared using the Oxford Nanopore Sequencing kit (SQK-MAP006 for R7 or SQK-NSK007 for R9), the Rapid library prep kit (SQK-RAD001), or the 1D ligation library prep kit SQK-LSK108. A 0.4× (instead of 1×) ampure cleanup was introduced after the FFPE DNA repair and the end-repair steps in the protocol to ensure removal of small DNA fragments. Genomic libraries were sequenced on R7.3, R9 and R9.4 flowcells followed by base calling using either Metrichor workflows or MinKnow software. For Patient 2, we introduced a DNA size selection step prior to library preparation using the Pippin HT system (Sage Science).

**Illumina whole-genome sequencing.** Genomic DNA of the patients and the parents was sheared to 400–500 bp fragments using the Covaris. Subsequently, genomic libraries were prepared using the nano library preparation kit. Genomic libraries were sequenced on an Illumina HiSeq X instrument to a mean coverage depth of ~30×.

**Nanopore data mapping.** FASTQ files were extracted from base-called MinION sequencing data using Poretools (version 0.6.0)[50]. Subsequently, fastq files were used as input for mapping by LAST (version 744)[17], against the GRCh37 human

reference genome. Prior to mapping the full data set, we used the last-train function to optimize alignment scoring parameters using a sample of 1200 nanopore reads. Nanopore sequencing data were also mapped using BWA-MEM with the -x ont2d option, as required by Lumpy and Sniffles. MinION 2D runs can produce 2D sequence reads, i.e., data where both forward and reverse reads of a DNA duplex are collapsed into a single sequence read, which produce three sequences in a fastq file, termed 1D template, 1D complement, and 2D. Therefore, we filtered the LAST and BWA BAM files by only retaining one of these three "versions" for each read based on the following order of preference: 2D > 1D template > 1D complement.

**Illumina data mapping.** Illumina HiSeq X ten data were mapped to the reference genome using BWA-0.7.5a using "BWA-MEM -t 12 -c 100 -M -R". Reads were realigned using GATK IndelRealigner[51] and deduplication was performed using Sambamba markdup[52]. Short indels and SNPs were genotyped using GATK HaplotypeCaller, jointly for the Patient 1 trio and individually for Patient 2.

**Analysis of MinION sequencing error rates.** We generated a set of 1,064,470 random positions on chromosome 1 and excluded sites that were regarded as polymorphic based on Illumina GATK variant calling. For each of the remaining positions, the mismatch rate, deletion rate, and insertion rate were calculated using samtools mpileup. All positions were overlapped with a bed file consisting of homopolymers longer than or equal to 5 bp. Additionally, we retrieved the simple repeat track from the UCSC table browser for overlapping all genomic positions with simple repeats. GC content was calculated using a window size of 10 bp surrounding each genomic position.

**NanoSV algorithm.** The NanoSV algorithm developed here (https://github.com/mroosmalen/nanosv) uses LAST BAM files as input. We did not use BWA-MEM alignments as NanoSV input, because the reads are not always split in non-overlapping segments. More precisely, we observed that the following two (over-simplified) CIGAR strings may be produced, for two aligned segments originating from the same sequencing read: 6M4S and 4S6M, respectively. While at least some of these alignments are marked as secondary by BWA and can be simply discarded from the analysis, we found that the LAST alignment splitting of the same reads leads, in some cases, to identification of otherwise high-confidence structural variants. This observation was not further investigated for the purpose of this project. See Supplementary Fig. 25 for a real example extracted from our data.

NanoSV uses clustering of split reads to identify SV breakpoint junctions. In a first step, all mapped segments of each split read are ordered based on their positions within the originally sequenced read. The aligned read may contain gaps between its aligned segments, i.e., parts of the read that do not align anywhere on the reference genome, for example, due to insertions (Supplementary Figs. 10 and 11) or simply due to low-quality sequencing.

Let tuple $x = (c,s,e,k)$ describe an aligned sequence segment, where the chromosome and genomic start and end coordinates of the segment are specified by $c$, $s$ and $e$, respectively, and the mapping orientation by $k \in \{+,-\}$. The coordinates $s$ and $e$ represent the start (lowest) and end (highest) coordinate of the mapped segment on the reference genome. Now, read $R_i$ can be described in terms of the ordered list of aligned segments and alignment gaps $X_i = [u_1, x_1, u_2, x_2, u_3, \ldots, x_N, u_{N+1}]$, where the ordering is determined based on their occurrence in the read, $u$ is the gap (i.e., unaligned sequence preceding segment $x$) and $N$ is the total number of aligned segments for read $R$. Alignment gaps are defined as read segments that are either unaligned or segments that fail to reach the mapping quality threshold $Q_1$ (default: 20). The size of an unaligned segment is denoted as $|u|$, and can be zero in case two adjacent segments align successfully.

Any two consecutive aligned segments $[x_n, u_n, x_{n+1}]$ in a read define a candidate breakpoint junction.

We further aggregate evidence from different reads supporting the same candidate breakpoint junction. This is achieved by clustering all candidate breakpoint junctions that have the same orientation and have start and end coordinates that are in close genomic proximity. In order to facilitate clustering of reads that cover the same breakpoint junction but that map to opposite strands of the reference human genome, order and orientation of the aligned segments is reverse complemented if for the genomic coordinates {p,q} mapping to the two closest bases of segments $x_n$ and $x_{n+1}$, respectively, within a given sequence read $R_x$, at least one of the following conditions is met:

1.  $p$ and $q$ are on the same chromosome and $p-q > 0$
2.  $p$ and $q$ are on different chromosomes and $p$ has a higher chromosome number

The clustering is initialized by assigning each pair of consecutive aligned segments $[x_n, u_n, x_{n+1}]$ to a separate cluster. The resulting clusters are then recursively merged. Any two clusters ($C_x$ and $C_y$) are merged if and only if, there exists a candidate breakpoint junction tuple $(x_n, x_{n+1}) \in$ cluster $C_x$ and a candidate

breakpoint junction tuple $(y_m, y_{m+1}) \in$ cluster $C_y$, such that the following conditions are met:

$$x_n(c) = y_m(c)$$
(segments $n$, $m$ map to same chromosome)
$$x_{n+1}(c) = y_{m+1}(c)$$
(segments $n+1$, $m+1$ map to same chromosome)
$$x_n(k) = y_m(k)$$
(segments $n$, $m$ have same orientation)
$$x_{n+1}(k) = y_{m+1}(k)$$
(segments $n+1$, $m+1$ have same orientation)
$$\min_{x,y}(|x_n(e) - y_m(e)|) \leq d$$
if $x_n(k) = +$ ($n$, $m$ segment_ends are in close proximity)
$$\min_{x,y}(|x_n(s) - y_m(s)|) \leq d$$
if $x_n(k) = -$ ($n$, $m$ segment_starts are in close proximity)
$$\min_{x,y}(|x_{n+1}(s) - y_{m+1}(s)|) \leq d$$
if $x_{n+1}(k) = +$ ($n+1$, $m+1$ segment_starts are in close proximity)
$$\min_{x,y}(|x_{n+1}(e) - y_{m+1}(e)|) \leq d$$
if $x_{n+1}(k) = -$ ($n+1$, $m+1$ segment_ends are in close proximity)

Where $d$ is the threshold that we set for the maximum distance between the alignment coordinates of two segments if they are to be considered as supporting the same breakpoint junction (default: 10 bp). Iterative clustering continues until no more clusters can be merged. Each final cluster represents one candidate SV, which is described by tuple $b = (c_1, c_2, p_1, p_2, k_1, k_2, g)$, with $p_1$, $p_2$ the medians of the start and end coordinates of all candidate breakpoint junctions contained in the cluster, $c_1$, $c_2$ the chromosomes associated to these coordinates and $k_1$, $k_2$ the orientation of the breakpoint junction. Finally, the gap size $g$ denotes the median size of the unaligned segments $u_n$, between the two consecutive aligned segments $x_n$ and $x_{n+1}$ of all the tuples within the respective cluster.

A true SV is called when a candidate SV is supported by more than $T$ reads (default: 2). Moreover, SVs with median mapping quality of the supporting reads not exceeding $Q_2$ are still reported, but flagged as "MapQual" in the VCF FILTER field. SV types can be determined from tuple $b$. Breakpoint junctions, where $c_1$ and $c_2$ point to different chromosomes, are considered interchromosomal SVs (e.g., chromosomal translocations), which can have one of four possible orientations (3′–3′, 3′–5′, 5′–5′, 5′–3′). Similarly, breakpoint junctions, where $c_1$ and $c_2$ point to the same chromosome, are intrachromosomal SVs, which can have one of four possible orientations (inversion type = 3′–3′ or 5′–5′, deletion/insertion type = 3′–5′, tandem duplication type = 5′–3′). Insertions and deletions are discerned based on the relation between the gap size, $g$, and the reference length $l = |p_1 - p_2|$, where an insertion is called if $g > l$ and a deletion is called when $g <= l$ (Supplementary Fig. 11).

We only consider two possible alleles for each SV candidate (present = ALT/ absent = REF). The reads supporting the alternative allele contain the segments constituting the breakpoint junction cluster. We consider as supportive of the reference allele all reads for which there is an aligned segment crossing one of the ends of the breakpoint junction (or both). More formally, a read is defined as crossing a breakpoint if it contains at least one aligned segment $x_n$ for which holds: $(p_1 - x_n(s) > 100 \land x_n(e) - p_1 > 100) \lor (p_2 - x_n(s) > 100 \lor x_n(e) - p_2 > 100)$. Reads not supporting the reference allele according to this definition are ignored. SV genotypes (homozygous alternative, heterozygous, homozygous reference, not-called) are assigned based on a Bayesian likelihood similar to the one used (and formally defined) by the SVTyper[23]. SV calls are reported in VCF format following the VCF standards as maintained by samtools specifications[53]. To facilitate reporting of complex SV types, such as inversions or reciprocal translocations, individual breakpoint junctions that bridge the same chromosomal regions, but are opposite in orientation (e.g., 3′–3′ and 5′–5′ for inversions), are linked using an identifier.

**Nanopore data SV calling.** We run NanoSV on the MinION data of each patient using the default parameters: "-t 8 -s 10 -p 0.70 -m 20 -d 10 -c 2 -f 100 -u 20 -r 300 -w 1000 -n 2 -q 80 -i 0.80 -g 100 -y 20". We discarded all sites where the alternative allele count was 0 in the resulting genotype (i.e., HOM_REF) and further filtered the resulting call sets for SVs tagged as "Cluster". The "Cluster" VCF INFO-field tag is added to all SV calls, which lie inside a (default) 1000 bp region containing three SVs or more. These clusters of SVs are most likely either sequencing errors or located in highly repetitive and/or decoy regions of the human reference. We used Lumpy[23] and Sniffles[24] (specifically designed for ONT and PB data) to call SVs in both samples using BWA-MEM alignments (instead of LAST alignment, as per requirement of the respective callers) of the same data and settings that match our own (liberal) NanoSV settings as closely as possible, as follows. For Lumpy: "-mw 2 -tt 0 -e", requiring that at least one read supports each candidate breakpoint and clustering breakpoints within 10 bp (back_distance = 10). For Sniffles: "-s 2 --max_num_splits 10 -c 0 -d 10"[24]. At the time of our analysis, SVTyper was not supporting nanopore reads (i.e., it required paired-end reads), therefore we

considered the Lumpy, ungenotyped, SV candidate sites as final calls for all subsequent analyses/comparisons. This implies that all sensitivity estimates for Lumpy are upper bounds and that precision estimates are most likely underestimated.

**Simulation of nanopore reads containing structural variants.** We took the human reference chromosome 1 sequence (GRCh37) and introduced 501 breakpoints, followed by random reshuffling of chromosomal segments into a new sequence. The breakpoints were introduced in a 20MB region (chr1:51707947–71707947), similar in size as a typical chromothripsis region. Subsequently, NanoSim[25] was used to simulate nanopore reads. We used 400 random reads from Patient 2 to build the error profiles for the simulated reads. Simulated read sets were generated for both the reshuffled chromosome 1 and the reference chromosome 1, in order to be able to introduce heterozygous structural variations in the simulated read data. Simulated reads were mapped using LAST and BWA, followed by SV calling using NanoSV, Lumpy and Sniffles, as described above. We performed downsampling of the reads to evaluate the effect of coverage on simulated breakpoint detection. Four of the randomly generated SV breakpoints produced small events (~40–50 bp), for which the LAST alignment does not result in a split read; these events were missed by NanoSV, regardless of the coverage used.

**Random forest variant filtering model.** We trained a random forest (RF) model that we subsequently used to filter out false positive SV calls from our nanopore data, such that we obtained a high precision set of variants for downstream analysis. The training data for our model consists of 354 true positive (TPs) SVs and 300 false positives (FPs). These 654 training data SVs are the NA12878 SV genotypes described in "Results" section, where any SV overlapping any of 1KG or PB data sets is considered a TP and all other SVs are considered FPs. Our training data are conservative in the sense that while all SVs considered TPs are based on previously curated data, we denote false positive SVs solely by overlap with other (different data) data sets (i.e., we performed no validation on NA12878 to evaluate if all/most novel variants that we find are indeed FPs).

We supply the following features to the RF model (where side 1 and side 2 refer to the lowest and highest genomic coordinates of a breakpoint junction, respectively; and the mean decrease Gini for each feature—proportional to the efficiency of splits in the model based on the respective feature—following in bold):

- **Mapq1:** average mapping quality over all reads supporting side 1 of the breakpoint junction (**5.78**)
- **Mapq2:** average mapping quality over all reads supporting side 2 of the breakpoint junction (**4.39**)
- **Pid1:** average percent identity (i.e., to the reference) over all reads supporting side 1 of the breakpoint junction (**27.10**)
- **Pid2:** average percent identity (i.e., to the reference) over all reads supporting side 2 of the breakpoint junction (**31.73**)
- **Cipos1:** genomic distance from the median start position of the SV to the lower bound of its associated confidence interval (**21.22**)
- **Cipos2:** genomic distance from the median start position of the SV to the upper bound of its associated confidence interval (i.e., confidence interval width = cipos1 + cipos2) (**17.08**)
- **Plength1:** average proportion of the aligned segment (i.e., relative to the entire read length), across all segments supporting side 1 of the breakpoint junction (**32.02**)
- **Plength2:** average proportion of the aligned segment (i.e., relative to the entire read length), across all segments supporting side 2 of the breakpoint junction (**43.44**)
- **Ciend1:** genomic distance from the (median) end position of the SV to the lower bound of its associated confidence interval (**17.73**)
- **Ciend2:** genomic distance from the (median) end position of the SV to the upper bound of its associated confidence interval (i.e., confidence interval width = ciend1 + ciend2) (**26.72**)
- **TotalCovNorm:** depth coverage summed across both ends of the breakpoint junction, divided by the average depth of coverage across the sample (**13.44**)
- **Vaf:** percentage of the reads spanning either end of the breakpoint junction that support the variant allele (i.e., the presence of a breakpoint junction) (**82.81**)

We found that most of our selected features were statistically significantly different across the sets of TPs and FPs, respectively (Supplementary Fig. 26), thus informative to our model.

The precision-recall curve of the model, and its 95% confidence interval, displayed in Supplementary Fig. 19 is derived from 100 bootstrapping runs, where the whole training data were split into 90–10% train-test subsets. The optimal operating point was chosen at 82% precision and 75% recall, and the final model used was trained again using all the training data available.

We compared the distributions of the random forest features used, across the training data of NA12878 and the test data of Patient 1 and Patient 2, respectively, and observed no statistically significant difference (Supplementary Fig. 27), We note that some difference should in fact be expected in the Patient 1 comparison, in the average read percent identity related features used (pid1 and pid2), given the different chemistries and nanopores used to generated these data.

**Illumina data SV calling.** SV calling for Illumina data was done using Manta[20], Delly[19], FREEC[34], Mobster[33], and Pindel[35]. For Manta we used version 0.29.5 with standard settings, for Delly we used version 0.7.2 with "-q 1 -s 9 -m 13 -u 5", for FREEC we used version 7.2 with window = 1000, for Mobster we used version 0.1.6 with standard settings (Mobster properties template), for Pindel we used version v0.2.5b8 with standard settings and excluding regions represented by the UCSC GRCh37 gap table using the -c option. Homozygous reference calls (genotype = 0/0) were omitted from the call sets for each of these tools.

**PCR, primer design, and SV validations.** Primers for breakpoint junction validation were designed using Primer3 software[54]. Breakpoint junction coordinates and orientations were used as input for primer design. Amplicon sizes varied between 500 and 1000 bp. PCR reactions were performed using AmpliTaq gold (Thermo Scientific) under standard cycling conditions. PCR products were sequenced using MiSeq (TruSeq library preparation, Illumina), Sanger sequencing (Macrogen), or MinION sequencing (2D library preparation and sequencing).

We performed extensive and heterogeneous validation on the SV calls of Patient 1, in order to obtain a thorough and reliable characterization of our data set and an informative comparison to standard approaches. We first randomly selected 384 NanoSV candidate calls (uniformly distributed across the observed size-range of SVs) from the call set of Patient 1 and performed validation with Illumina MiSeq. We further selected 400 candidate calls (uniformly distributed across the observed size-range of SVs) exclusively from the nanopore-specific SV calls and validated them. Deep coverage MinION sequencing was used for this second round of validation, under the assumption that a long-read accessible-only set of variants would be less likely to validate using the short-read Illumina sequencing. A third round of validation was performed, also by MinION deep coverage sequencing, on a set of 192 non-random variants; namely, 96 variants were expected to be true positive SV calls and 96 false positive SV calls, as of an initial attempt to build a discriminative model. Upon inspection of these validation results, SVs falling within homopolymer stretches (see above) and/or short tandem repeats (UCSC tandem repeat table) were considered unreliably genotyped (i.e., even in the validation data) and were subsequently discarded from the data set altogether (see main text—"Results" section).

All of the above three rounds of validation are thus restricted to the sites that fall outside homopolymers and/or short tandem repeats and SVs for which we did not obtain a specific PCR product are discarded. This is the subset that is referred to as validation data throughout the text, when evaluating precision and it consists of 274 SVs (185 true positives and 89 false positives).

Finally, we selected 14 large inverted breakpoint junctions (>1000 bp) plus 82 randomly selected SV candidates, all of which were predicted as true by the random forest model from Patient 2. We performed PCR for each of these 96 SV breakpoint junctions and sequenced the resulting amplicons using deep coverage MinION sequencing. We were able to successfully produce and sequence amplicons for 77 of the variants, and 74 of them validated. Out of the 14 large inverted breakpoint junctions, eight produced a PCR product and seven of these were validated as true.

A structural variant was considered validated as a true positive if there exists an SV call, in the validation SV call set, that overlaps (in the meaning described below) the original SV validation candidate. The validation SV call set is produced similarly to the initial analysis, where Manta is used for genotyping SVs in the MiSeq validation data and NanoSV is used for the nanopore data, respectively, with the note that deep coverage (i.e., ~1000 for MiSeq and MinION runs) enables accurate genotyping.

**Annotation of repeat elements.** All deletions from our NanoSV call set were annotated as overlapping a repeat element, if the sequence of the variant overlaps an entry of the repeat masker table of UCSC (GRCh37). For all NanoSV variants reported as insertions, we extracted the inserted sequence as identified in each supporting nanopore read, used Muscle[55] to generate a multiple sequence alignment for all the sequences supporting the same insertion and obtained a consensus sequence by a simple majority vote. Subsequently, we interrogated the DFAM[56] database and annotated all insertions which contained sequence of a repeat element.

**Calculating overlap between SV data sets.** To calculate the intersection between SV call sets, we considered each SV call as a set of breakpoint junction start and end coordinates $s$ and $e$, and orientation $k$. For any SV call $i$, each breakpoint junction coordinate ($s_i$ and $e_i$) is the median of an associated confidence interval, ($s_{i,l}, s_{i,h}$) and ($e_{i,l}, e_{i,h}$), respectively, as derived from the evidence cluster $C_i$. SV calls $i$ and $j$ are overlapping if the confidence intervals of their start and end coordinates are closer together than 101 bp. For SVs smaller than 1000 bp (excluding insertions), we additionally required that SVs overlap each other with a reciprocal overlap of at least 70%, otherwise, considering the 100 bp margin that we use when comparing breakpoint junction borders, different SVs that happen to be in genomic close proximity may, incorrectly, be considered the same event.

**GC bias.** The GC content (i.e., percentage of guanine or cytosine bases within a certain DNA sequence) was computed for 100,000 5 kb intervals of the reference genome (build GRCH37). These intervals were chosen such that they do not overlap sequencing gaps in the reference, as defined in the UCSC table browser, including telomers, centromeres, and other gaps. The average depth of coverage across each interval was then computed from the HiSeq alignment data and the MinION alignment data, respectively (stratified by sequence reads tagged as "passed" and "failed" by the Metrichor base calling for Patient 1). The GC content was binned into 30 uniformly spread bins, between the minimum and the maximum GC content derived from the data. Six GC content bins were discarded—i.e., those where GC content <0.26 or GC content >0.66—as too few sampled intervals fell within these bins and a coverage distribution cannot be robustly estimated (i.e., 1–18 intervals per bin, Supplementary Fig. 6).

A linear regression model with average coverage as the dependent variable and GC content as the independent variable was fitted, in order to quantify the GC bias of the two sequencing technologies, respectively. The average coverage values were normalized (0 mean, 1 variance) for Illumina and MinION data, respectively, because of the different sequencing average depth of coverage, such that the regression coefficients for the two technologies be comparable (i.e., the resulting regression coefficients express the number of standard deviations that the coverage varies, per GC content percentage).

**Genetic phasing of variants from Illumina sequencing data.** We used the Illumina whole-genome sequencing data of Patient 1 and both its parents to obtain a high-confidence set of phased genotypes (including SNVs, short indels, and SVs), against which we subsequently evaluated the nanopore data analysis. We used GATK PhaseByTransmission (PBT)[27] to correct genotypes based on trio information and to obtain deterministic phasing for most loci. PBT settings were: "-prior 0.000001 -useAF GT -af_cap 0.0001". The PBT-phased SNVs were used to evaluate the genome-wide read-backed phasing from nanopore data as well as for phasing the nanopore reads and the PBT-phased SVs were used to evaluate the nanopore read-backed phasing of the SVs (i.e., evaluation was limited to the SVs detected in both nanopore and Illumina data). PBT was run with a de novo mutation prior of 10e-6 and supplied with the population allele frequencies of 1KG phase 3 European population.

**Nanopore read-based phasing of SNVs using WhatsHap.** For both patients, all bi-allelic heterozygous SNVs were phased from the aligned MinION reads using WhatsHap[40,41] (version 0.13 + 21.g45bd7f8, command line "whatshap phase--distrust-genotypes--reference <ref.fasta> <variants.vcf> <reads.bam>"), i.e., with realignment mode enabled. That is, reads were realigned against reference and alternative alleles at variant sites, which is critical for phasing performance of noisy long reads[41]. For comparison purposes, we used SNV genotypes to obtain a population-based phasing with respect to the 1KG phase 3[33] reference panel by running ShapeIt with default parameters. We excluded from the comparison all variants that fell within homopolymer runs longer or equal to 5 bp, due to both genotyping accuracy, but mostly because of the known drop in sequencing accuracy of MinION reads for longer homopolymer sequences. The homopolymer bed-track used was computed genome-wide, incorporating a 1 bp border around the homopolymer, such that relatively frequent sequences of the form "XXXXXYZZZZZ" be merged into one homopolymer region for the final result.

**Phasing of nanopore reads and SVs.** Individual nanopore reads from Patient 1 were phased using a set of 1.7M heterozygous SNVs that were genetically phased by GATK PBT[27]. Individual nanopore reads were phased using the genetically phased SNVs by determining the base call and corresponding base quality at each SNV position within each read. Let $b(i)$ and $q(i)$ be the base call and associated quality value for some SNV $i$ in some read under evaluation. Further let $BM(i)$ and $BP(i)$ be the maternal and paternal alleles, respectively (i.e., as phased by PBT), for SNV $i$. The information from all SNVs spanned by a read is then aggregated and the likelihood that read $r$ originates from the paternal or the maternal haplotype, respectively, is computed:

$$L_p(r) = \prod_{i=1}^{n} P(b(i)|\text{BP}(i))$$

$$L_m(r) = \prod_{i=1}^{n} P(b(i)|\text{BM}(i))$$

Where $n$ is the total number of SNVs that read $r$ overlaps and

$$P(b(i)|\text{base}) = 1 - 10^{\frac{-q(i)}{10}}, \text{ if } b(i) = \text{base}$$

$$P(b(i)|\text{base}) = 10^{\frac{-q(i)}{10}}, \text{ if } b(i) \neq \text{base}$$

Is the probability that a read supports a specific phased allele at an SNV. The likelihoods that the SV resides on the paternal or the maternal haplotype,

respectively, are then computed:

$$L_p(\text{SV}) = \prod_{r=1}^{R_{\text{SV}}} L_p(r)$$

$$L_m(\text{SV}) = \prod_{r=1}^{R_{\text{SV}}} L_m(r)$$

Where $R_{sv}$ denotes the set of all reads supporting the breakpoint junction. The two likelihood scores are then transformed to probabilities (i.e., normalized to sum up to 1) and phase for the set of breakpoint junction supporting reads is assigned as indicated by the highest likelihood score. Phase is assigned identically to the set of reference supporting reads spanning the breakpoint junction.

An SV is then considered phased if the two phases, for the set of breakpoint supporting reads and reference supporting reads, respectively, correspond to different parental haplotypes and the (phred scaled) phasing posterior quality is defined as:

$$\text{PP} = -10 \times \log_{10}\left(\max\left(L_p(\text{SV}), L_m(\text{SV})\right) \times \max\left(L_p(\text{REF}), L_m(\text{REF})\right)\right)$$

**Reconstructing the chromothripsis from alignment overlaps.** To obtain evidence for the long-range structure of the chromothripsis breakpoint junctions in Patient 1, we first extracted the set of (aligned) nanopore reads that span the chromothripsis regions on chromosomes 1, 7, 8, and 9. Separation of reads by phase was done as described above. The mapped segments were ordered by left genomic mapping coordinate of each segment to produce an ordered list of segments $L=\{s(1), s(2), \ldots, s(n)\}$, from all reads jointly. We then built an oriented graph, where each aligned segment in $L$ is initially a node and nodes were iteratively merged as follows: Let $i$ and $j$ represent the start (left) and end (right) coordinate of each segment (i.e., or, subsequently, nodes). In order for $s(x)$ to be merged into a node $s(y)$, there must exist at least two other segments $s(z)$ and $s(t)$ supporting the same node, such that $(s(z)_j - s(x)_i) >= 20$ bp and $(s(t)_j - s(x)_i) >= 20$ bp. Edges were then added, to the final, reduced set of nodes, by evaluating each read's segmentation across supported nodes. Namely, an edge is added between any two nodes for which there is a read such that one segment of the read supports one node and another segment of the same read supports the other node. Each edge was then weighted by the number of reads supporting that connection.

Finally, contigs were built by evaluating all maximal length paths through the graph, where only edges of weight at least two are considered and branching is resolved in a greedy way, by selecting the maximum weight path at each step.

Using the above algorithm, individual breakpoint junctions were connected together, providing support for the order of the joined segments in the chromothripsis chromosomes of Patient 1.

**Assembly of MinION sequencing data.** Nanopore reads of Patient 1 were separated into three bins by phase, as described above. The reads that were assigned a paternal phase and the unphased reads were used as input for de novo assembly using Miniasm[28], with settings: minimap -S -w 5 -L 100 -r 500 -m 0 and miniasm -c 1 -m 100 -h 20000 -s 100 -r 1,0 -F 1. The mentioned parameters were found to produce the longest contigs from our data. These Miniasm contigs were subsequently aligned to the human reference genome (GRCh37) using LAST, with settings: -s 2 -T 0 -Q 0 -p [last_parameters]. The last_parameters were obtained as described above (i.e., the same used for aligning the initial MinION data of Patient 1 and Patient 2). LAST aligments (SAM format) were processed by custom scripts to evaluate the presence of chromothripsis segments from Patient 1 based on chromosomal coordinate overlap.

**Data availability.** Illumina and nanopore whole-genome sequencing data used in this study can be accessed through the European Genome-phenome Archive under accession number EGAS00001002333. All other relevant data are available from the corresponding author on request.

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

## Acknowledgements

We thank the Bioinformatics Expertise Core of the UMC Utrecht for setting up part of the computational infrastructure and software to analyse nanopore sequencing data. This work was supported by funds from the Utrecht University to implement a single-molecule sequencing facility. We thank Paul I.W. de Bakker for supporting M.C.S. from VIDI Grant 91712354 from the Dutch organization for scientific research (NWO-ZONMW). M.E.T. was supported by grants from the National Institutes of Health (GM061354 and HD081256). G.P. is supported by a fellowship of the Associazione San Luigi Gonzaga. We thank Eleonora Di Gregorio, Alfredo Brusco, and Elisa Savin for their contribution to the identification of the complex chromosomal rearrangement in Patient 1.

## Author contributions

S.M., G.P., D.G., G.M., J.K., and M.E.T. provided access to patient cells and DNA. I.R. and W.P.K. generated MinION sequencing data. E.d.B., J.K., and E.C. provided Illumina sequencing data. M.C.S., M.J.v.R., M.M.N., J.d.L., J.E.V.-I., T.M., J.d.R., and W.P.K. performed nanopore data analysis. M.C.S., W.P.K., T.M., and J.d.R. wrote the manuscript. All authors contributed to the final version of the manuscript.
