## [Peer Review File · Nature Communications]

Reviewers' Comments:

Reviewer #1:

Remarks to the Author:

I have seen a previous version of this manuscript, which describes a timely and very relevant set of analyses of structural variation using Oxford Nanopore sequencing -- especially, the sequencing of two patient

genomes with congenital abnormalities using the ONT MinION, at 11x and 16x coverage respectively, as well as a bioinformatic pipeline - NanoSV - to efficiently map SVs from the long-read data. The authors show that their approach facilitates detecting complex SVs and also that many previously missed SVs can be identified in the genome using their approach.

The authors have adequately addressed the criticisms made in relation to the previous version of this manuscript. I would recommend acceptance of this timely piece.

Reviewer #2:

Remarks to the Author:

The authors have done a good job to address the reviews. Their work is important, and the paper may help further the uptake of this sequencing platform. They do have one speculative comment in the discussion about 'long reads being the future'...hard to say right now (kindof like electric cars). I would suggest removing this.

Reviewer #3:

Remarks to the Author:

Cretu Stancu, van Roosmalen et al have revised their manuscript and included new types of analyses. These make the manuscript much stronger than its first version. Especially the NA12878 analyses add value to the manuscript. Most other points of mine have also been taken into account, so that the manuscript should be more easily understandable. From a scientific point of view, I therefore now fully support publication in this journal.

There are still one or two places where the grammar appears weird to me. I would suggest having a native speaker read this in detail. This does not require further review.

REVIEWERS' COMMENTS:

Reviewer #1 (Remarks to the Author):

I have seen a previous version of this manuscript, which describes a timely and very relevant set of analyses of structural variation using Oxford Nanopore sequencing -- especially, the sequencing of two patient

genomes with congenital abnormalities using the ONT MinION, at 11x and 16x coverage respectively, as well as a bioinformatic pipeline - NanoSV - to efficiently map SVs from the long-read data. The authors show that their approach facilitates detecting complex SVs and also that many previously missed SVs can be identified in the genome using their approach.

The authors have adequately addressed the criticisms made in relation to the previous version of this manuscript. I would recommend acceptance of this timely piece.

We thank the reviewer for the positive response to our revised manuscript.

Reviewer #2 (Remarks to the Author):

The authors have done a good job to address the reviews. Their work is important, and the paper may help further the uptake of this sequencing platform. They do have one speculative comment in the discussion about 'long reads being the future'...hard to say right now (kindof like electric cars). I would suggest removing this.

We thank the reviewer for the positive response to our revised manuscript.

We have adapted the respective sentence to reflect the uncertainty of such speculations/predictions (Page 12).

Reviewer #3 (Remarks to the Author):

Cretu Stancu, van Roosmalen et al have revised their manuscript and included new types of analyses. These make the manuscript much stronger than its first version. Especially the NA12878 analyses add value to the manuscript. Most other points of mine have also been taken into account, so that the manuscript should be more easily understandable. From a scientific point of view, I therefore now fully support publication in this journal.

There are still one or two places where the grammar appears weird to me. I would suggest having a native speaker read this in detail. This does not require further review.

We thank the reviewer for the positive response to our revised manuscript. We took a more critical look at various formulations. Additionally, the main text was read and corrected by a native speaker.